# Artificial Urinary Sphincters for Moderate Post-Prostatectomy Incontinence: Current Research and Proposed Approach

**DOI:** 10.3390/cancers15184424

**Published:** 2023-09-05

**Authors:** Andrew Johnson, Spencer Mossack, Peter Tsambarlis

**Affiliations:** 1Rush Medical College, Rush University, Chicago, IL 60612, USA; andrew_d_johnson@rush.edu; 2Department of Urology, Rush University Medical Center, Chicago, IL 60612, USA; spencer_m_mossack@rush.edu

**Keywords:** post-prostatectomy incontinence, continence surgery, artificial urinary sphincter, male sling

## Abstract

**Simple Summary:**

Stress urinary incontinence is a common sequela in men after radical prostatectomy for the treatment of prostate cancer. Varying degrees of post-prostatectomy incontinence will present after surgery and surgical treatment may be recommended after one year. The severity of incontinence can range from less than 1 urinary pads per day (PPD) to more than 5 PPD. Treatments for mild incontinence include the male sling while more severe incontinence often requires an artificial urinary sphincter (AUS). Currently, patients with moderate incontinence are treated with either a sling or AUS with variable results. In this paper, we reviewed recent research to demonstrate that AUS should be considered first-line for moderate incontinence. While patients and physicians may be hesitant to proceed with an implantable device, patients achieved better continence rates and overall quality of life when they underwent AUS placement for moderate post-prostatectomy incontinence.

**Abstract:**

Male urinary incontinence is a common complication after radical prostatectomy. The severity of incontinence can be assessed in various ways and helps determine the best surgical intervention to restore continence. While most patients with mild incontinence receive a sling and those with severe incontinence receive an artificial urinary sphincter (AUS), there are no clear guidelines on how to manage patients with moderate post-prostatectomy incontinence (PPI). Our discussion will focus on the current literature, which demonstrates that an AUS should be considered first-line in men with moderate PPI despite perceived concerns over complications and reintervention rates.

## 1. Introduction

Male urinary incontinence is one of the more devastating, yet common, complications after radical prostatectomy for prostate cancer. Post-prostatectomy incontinence (PPI) can range anywhere from 4 to 69%, but these rates vary significantly based on the severity of symptoms. Severe incontinence occurs in approximately 8% of patients after prostatectomy [1,2,3]. Currently, the AUA guidelines recommend considering continence surgery as early as 6 months post-prostatectomy in patients suffering from PPI resistant to conservative treatment. Furthermore, continence surgery is recommended after 1 year as only an additional 1% of patients will show improvement in incontinence between 12 and 24 months post-operatively [4]. The two most common surgical treatments for PPI are the artificial urinary sphincter (AUS) and the male sling. The male sling is a mesh material that is placed near the bulbar urethra that helps reposition the rhabdosphincter or urethra to help patients achieve continence. In contrast, the AUS is a balloon device that sits around the urethra and can be manually inflated (to prevent leakage) or deflated (to allow voiding) by the patient. For over 30 years, the AUS has proven to be a reliable treatment option for PPI; however, due to concerns over device infection, malfunction and the possible need for revision, the male sling has gained popularity as a less-invasive approach to PPI.

The decision about which surgery to pursue depends on the severity of incontinence as well as patient and surgeon preference. Consistent with societal guidelines, patients with mild incontinence will often receive a male sling and patients with severe incontinence will frequently receive an AUS [4]. Currently, for men with moderate stress urinary incontinence (SUI), which accounts for 40–53% of PPI patients, the choice of surgery will be based on patient and surgeon preference as a guideline-directed therapeutic protocol has not yet been established for this group [5,6]. While various complications can result after radical prostatectomy, urinary incontinence can have a severe impact on a patient’s quality of life [7]. Because there are no formal guidelines for moderate SUI, and nearly half of all patients with PPI fall into this moderate category, we believe it is important to assess recent studies comparing AUS to the male sling. In this review, we assess the literature comparing AUS to the male sling for men with moderate PPI and describe our approach to moderate PPI at our institution.

## 2. Methods and Inclusion Criteria

This article was developed as a non-systematic, narrative review that reflects our approach to moderate incontinence following prostatectomy at our institution. The following search strategy was utilized. PubMed was selected as the primary database for gathering studies. The following terms were searched: “post-prostatectomy incontinence (PPI)”, “artificial urinary sphincter”, “male sling”, “prostate cancer”, “membranous urethral length”, “prostate cancer radiation”, “stress urinary incontinence (SUI)”, “Modified Stress Incontinence Grading Scale (MSIGS)”, “patient hesitancy”, “urinary continence”. While few studies have been conducted comparing the efficacy of the male sling versus AUS, Table 1 below outlines those selected for this article. The current AUA guidelines on this subject matter were revised in 2019. The authors’ hope is to prioritize studies that have been published since 2019 that can be reasonably used to assess the treatment of moderate post-prostatectomy incontinence.

## 3. Risk Stratification

Once surgical treatment is appropriate for PPI, patient selection is critical to achieving optimal surgical outcomes. The current AUA guidelines advise against placing slings in men with severe PPI or men who have a history of pelvic radiation as studies have shown poor continence rates during follow-up. Although an AUS is recommended in this patient population, patients should be counseled on an increased risk of complications and lower continence rates compared to patients without a history of radiation. Providers should avoid placing an AUS in patients if there are concerns regarding cognitive or memory deficits, poor manual dexterity, any degree of decreased sensation, or complications related to overactive bladder. Most of the studies included in this review adhere to similar exclusion criteria. As treatment for prostate cancer can be approached from several different modalities, these limitations should be considered.

The measurement of preoperative membranous urethral length (MUL) via MRI has also been used to predict rates of return of continence post-prostatectomy. The membranous urethra, found between the prostatic and bulbar urethra, may provide clinicians with more data when assessing the severity of incontinence that can be expected after radical prostatectomy. In 2016, Mungovan conducted the first systematic review and meta-analysis that demonstrated MUL is a useful marker when assessing for PPI. In a review of nearly 9000 patients combining 12 selected studies, every additional millimeter of MUL correlated with improved return of continence at 1-, 3-, 6-, and 12-month follow-up. Each extra millimeter of MUL increased the odds of recovery between 5% and 15% (OR: 1.09, 95% CI: 1.05–1.15, *p* < 0.001) and every extra 10 mm increased the odds of recovery between 63% and 205% (OR: 2.37, 95% CI: 1.63–4.05) [12]. The authors recommend that all patients have MUL measured via T2-weighted coronal and sagittal imaging prior to radical prostatectomy to help inform patient expectations regarding the return of post-operative continence. Finally, clinicians can use MUL as a means to modify surgical techniques to better preserve MUL intraoperatively to drive better patient outcomes. While it may seem obvious, surgeons with more skill in this field will likely demonstrate better outcomes for their patients. The greater the experience of the surgeon or the higher the volume of these types of procedures at an institution, the higher the probability that PPI could be avoided. While PPI may be unavoidable in more elderly patients or those with complicated bladder conditions and comorbidities, individual surgeon skill plays a large role in outcomes when treating PPI.

## 4. Assessing Incontinence

Prior to proceeding with surgical intervention, it is essential that the severity of SUI be established. Patients often undergo diagnostic testing including cystoscopy, uroflowmetry and post-void residual volumes. The gold standard to assess stress urinary incontinence, however, is the 24 h pad weight test. This test correlates with surgical outcomes and has been shown to be superior to pads per day (PPD) at estimating the severity of SUI [13,14]. PPD can overestimate leakage, for example, if a man consistently changes his pads before they are saturated, which is common. Unfortunately, 24 h pad weight testing is burdensome to the patient and provider due to both logistics and cost. Thus, more often practitioners rely on PPD in clinical practice. Mild SUI is often defined as 0–2 PPD, moderate as 3–5 PPD, and severe as >5 PPD. These were the cutoffs used when assessing the data for this review.

Alternatively, an in-office standing cough test can act as a surrogate for the 24 h pad weight test. This test allows the clinician to stratify the severity of SUI into five grades based on the leakage pattern during a strong cough in the standing position with a full bladder. Morey found a strong correlation between the 24 h pad weight and the standing cough test in the pre- and post-operative assessment of patients that received a sling or AUS for PPI. In this single-surgeon, retrospective study, 104 patients (63% received a sling and 37% received an AUS) received a pre-operative standing cough test and were assessed a grade using the Modified Stress Incontinence Grading Scale (MSIGS; Grade 0 = leakage reported but not demonstrated; 1 = delayed drops; 2 = early drops without stream; 3 = delayed stream; 4 = early and persistent stream). They found that using the MSIGS for the assessment of the severity for SUI strongly correlated with average pad weights (Spearman coefficient ρ = 0.68). Additionally, patients in the sling group were considered to have mild–moderate SUI with an average PPD of 2.4 and an MSGIS of 0–2, while the AUS group had moderate–severe SUI with a PPD of 3.6 and an MSIGS of 3–4 [15,16,17]. This study demonstrated a rapid and accurate method by which stress incontinence in the office could be assessed without the cumbersome reality of pad counting and weighing.

Despite the efficiency and accuracy of the standing cough test, it is not without its limitations and can be influenced by body habitus and fluid status. Some have suggested utilizing a repeated pad test over multiple days to obtain concise objective data regarding PPI [18]. Furthermore, although there is no gold-standard questionnaire, there are many subjective questionnaires that can be used to further elicit PPI severity. It is essential that urologists include a combination of subjective and objective measures when assessing the severity of incontinence in their patients prior to proceeding with treatment.

## 5. Current Practice

Three of the most common therapies for PPI are pelvic floor physical therapy, male sling, and AUS. The AUA guidelines recommend (Moderate Recommendation) pelvic floor muscle training in the immediate post-operative period. Patients with PPI, despite physical therapy, should then be considered for surgery at 6 months (Conditional Recommendation) and offered surgery at 1 year (Strong Recommendation). When surgical treatment is considered, an AUS can be offered to all patients with PPI (Strong Recommendation), while male slings should be considered only in those with mild to moderate PPI (Moderate Recommendation) [4]. The EAU guidelines recommend similar strategies for the surgical treatment of PPI [19]. With both the AUS and sling as potential options for men with moderate PPI, the treatment approach is often a product of shared decision-making.

Despite the well-documented success of the AUS, patients may be concerned about the risks of device infection (8.5%), mechanical failure (2–14%) and the need for re-intervention (26%), which may drive patients towards a male sling [20]. Since its introduction in 2006, the male sling has gained popularity for mild PPI. Success rates for male slings at 1 year follow-up ranged from 86% for pad weights <100 g, 83% for pad weights 100–400 g, and 40% for pad weights >400 g, suggesting that its efficacy is optimal in mild PPI and deteriorates with more severe PPI [21]. Despite the inferior efficacy in severe PPI, 25% of patients will opt for a sling even when an AUS is recommended by the surgeon. Furthermore, when given a free choice, over 90% of patients will opt for a sling, highlighting fears over the complications and the need to control an implantable device. Patient hesitancy and values are further discussed in Section 12 below. The sling is frequently described as a “set it and forget it” approach, whereas the AUS requires patient interaction with the device.

## 6. Male Sling vs. Artificial Urinary Sphincters: Types and Procedures

There are various models in use for both male slings and AUS. As discussed in the subsequent sections, patient selection is critical in achieving successful outcomes. Male sling procedures often include the implantation of a mesh sling that helps reposition the rhabdosphincter or urethra to help patients achieve continence. Slings can function as either a fixed or adjustable device. Currently, adjustable slings are not available in the United States. As such, we did not include adjustable slings in our review as they are not accessible at our institution. Common fixed models employed in the United States include Boston Scientific’s AdVance XP sling. Outside the United States, common adjustable models include Coloplast’s Virtue sling and Agency for Medical Innovation’s (Austria) Adjustable Transobturator Male System (ATOMS). While adjustable devices have demonstrated good continence rates, the current research only includes studies comparing these devices against themselves without comparison to a control or to AUS [22,23,24]. For the purposes of this paper, we decided to only include studies that compared AUS versus sling, which only included fixed models.

In comparison, the AUS includes three connected components: a cuff, control pump, and pressure-regulating balloon. The cuff encircles the bulbar urethra, the control pump is placed within the scrotum, and the pressure-regulating balloon is placed in the sub-rectus space in the lower abdomen. Patients control the AUS by manipulating the pump to decrease the pressure on the urethra and allow urine to flow. The most commonly used model is Boston Scientific’s AMS 800. Since its release in the 1980s, the device has remained largely unchanged and is considered the gold standard.

## 7. Comparing AUS to Sling Outcomes

In 2020, Kourhi assessed the use of slings vs. AUS for the treatment of persistent mild, moderate, and severe stress urinary incontinence. They reviewed the post-operative outcomes of 267 patients (sling 202, AUS 65) from a single-surgeon database. Utilizing MSGIS, as previously described, the patients were pre-operatively stratified into mild (Grade 0–1), moderate (Grade 2–3), and severe (Grade 4) SUI. Treatment failure was defined as having more than 1 PPD or the need for a second procedure. This cohort included 179 patients with moderate SUI (sling 114, AUS 65). Despite patients in the AUS group having higher baseline PPD than the sling group (4.2 vs. 2.5, respectively), patients in the AUS group had a significantly higher success rate than the sling group (80% vs. 63%, respectively, *p* < 0.01) [8]. The sling group did include 16% of patients with a history of radiation, which the authors state would now exclude them from being considered for a sling. Ultimately, the authors concluded that an AUS should be the preferred treatment for patients with MSIGS ≥ 2.

In 2021, Sacco replicated the findings previously demonstrated by Kourhi’s group. This retrospective study focused on patient-reported outcomes after at least 1 year of follow-up in post-prostatectomy patients with moderate SUI (defined as 3–5 PPD). The cohort included 35 patients in each treatment group after implementing 1:1 propensity score matching to reduce selection bias. At 12-month follow-up, 33/35 (94.3%) patients in the AUS group were “much improved” versus 24/35 (68.6%) in the sling group. There was a fivefold higher likelihood that patients progressed from “unchanged” to “worse” SUI in the sling group. The AUS group also outperformed the sling group in objective measures including reduction in PPD from baseline (AUS −4 vs. sling −3) and reduction in 24 h pad weight from baseline (AUS −455 g vs. sling −290 g). The AUS group had superior outcomes compared to the sling group despite presenting with higher preoperative PPD, again demonstrating the higher efficacy of the AUS in more advanced, but still moderate, PPI. Patients in the sling group had a higher rate of overall complications as well (37% vs. 20%, respectively). There were three patients who had complications requiring additional surgery after AUS surgery; however, the overall intervention rates did not differ between the groups because 25.7% of patients in the sling group needed a second surgery for PPI compared to 2.9% in the AUS group. Lastly, there were no cases of mechanical failure identified during the 1-year follow-up period [9]. Although the risk of more severe complications cannot be ignored in this cohort, these results emphasize the importance of educating patients about the possible need for future surgical interventions after receiving a sling. While some urologists may be hesitant to proceed to AUS without attempting a male sling first, this study suggests that a first-line AUS approach to moderate SUI is clinically appropriate and reduces the subsequent surgical risks and costs associated with a salvage AUS.

In 2021, Abrams conducted a multicenter, noninferiority, randomized controlled trial (MASTER) comparing patient-reported urinary incontinence one year after an AUS or sling surgery for PPI. The study included 190 patients in each treatment group. Despite there being no exclusion criteria based on the severity of SUI, the average PPD in each group was 3 pads and the average 24 h pad weight was 256–267 g, which correlates with a diagnosis of moderate SUI. When assessing post-operative incontinence as anything other than no urinary leakage, both groups performed poorly, with an incontinence rate of 87% in the sling group and 84% in the AUS group. While the study demonstrated noninferiority on primary analysis, a further analysis of the survey results demonstrated that patients who underwent AUS had more severe preoperative incontinence symptoms at baseline than their male-sling counterparts and had a statistically significantly greater improvement in symptoms than those with a male sling after surgery. In response to the question “Overall, how much does leaking urine interfere with your everyday life?”, male sling patients experienced greater interference post-operatively than AUS patients, suggesting that an AUS provided a greater improvement in their day-to-day life even if their pre-operative symptoms were more severe than those who received a male sling. Patients in the AUS group also had a higher satisfaction rate compared to the sling group and this held true for patients with a baseline pad weight of both less than and greater than 250 g. The overall non-serious adverse events did not differ between the two groups; however, there was a higher number of patients in the AUS group (eleven) compared to the sling group (six) who had “serious adverse events” that ranged from catheterization prolonging hospital stays to device erosion. Lastly, men in the sling group had a significantly higher incidence of needing future SUI surgery (7.2%) compared to the AUS group (1.8%), with twelve men having an AUS placed and one man undergoing a second sling operation. There were three cases of mechanical failure in the AUS group requiring reintervention [10]. This study showed superior patient-reported outcomes after AUS placement regardless of the severity of pre-operative incontinence, while redemonstrating the high incidence of men failing a sling and requiring an AUS in the future.

The MASTER trial is the only randomized controlled trial comparing AUS to slings for PPI. Most of the published literature focuses on retrospective studies with single-surgeon data and small sample sizes, making it difficult to draw strong conclusions that affect clinical practice. The MASTER trial, however, is limited by patient-reported outcomes, rather than more objective data as a measure of treatment success. In 2022, Lin synthesized much of the current literature in the first systematic review and meta-analysis comparing AUS to slings for the treatment of moderate PPI utilizing objective measures of success. They reviewed five studies totaling 509 patients (295 for sling and 214 for AUS). They included studies on men who had less than or equal to 5 PPD at baseline with at least 1 year follow-up. Treatment success was defined as daily pad use of 1 PPD or less post-operatively. Men in the AUS group had a significantly higher success rate than the sling group (OR 0.57, CI 0.30–0.90). The overall complication rates were equivocal between the two treatment groups [11]. The authors concluded that the success rate of the AUS was higher than that of slings with comparable complication rates. While the MASTER trial was not included in their review, both studies came to similar conclusions regarding the superiority of the AUS compared to slings for PPI. Overviews of the studies therein can be found in Table 1.

## 8. Limitations

While there is an abundance of literature describing the treatment of PPI, there is a scarcity of prospective studies comparing the AUS and sling, limiting the strength of our conclusion that the AUS is superior to the sling for moderate PPI. Furthermore, many of the studies we reviewed were published after the AUA released their guidelines on incontinence after prostate cancer treatment in 2019 [4]. As such, notable variability among the reviewed studies should be noted. Most patients were assessed one year after prostatectomy, but some studies failed to definitively outline the follow-up period. Additionally, some of the studies included a very small number of patients that had either not undergone prostatectomy or had received pelvic radiation prior to their treatments. While moderate incontinence is commonly defined as 3–4 PPD, some studies included patients with >5 PPD, which would largely be considered severe PPI. There was also widespread use of subjective measures of improvement, which are certainly important, but objective parameters (such as PPD or MSIGS) may be more useful for comparing outcomes between the two surgical interventions. To convey a concise perspective, we assumed that MSIGS and PPD are close approximates to present our findings in a meaningful way, which is supported by the findings in Yi’s study from 2020 [17]. Through our experience, the relationship and conversations between the urologist and the patient often determine which line of treatment is pursued. Therefore, patient preference is often subject to physician bias. It is important that physicians discuss the pros and cons of the male sling versus AUS prior to prostate cancer treatment and lay out the recent research to help inform that choice. In our experience, very few patients elect to undergo a sling when they have a full understanding of its risks and benefits when compared to AUS. This includes a concise, but complete, review of the data reviewed here.

While the studies described above have demonstrated a reasonable argument for AUS use in moderate post-prostatectomy incontinence, the field could benefit from high-quality, prospective randomized controlled trials (RCT) that compare AUS to male slings. Currently, the AUA guidelines recommend that patients be informed of the high rate of urinary incontinence after radiation therapy and that AUS be considered after persistent SUI after sling [4]. The aforementioned studies have produced insight into the differences between outcomes, but prospective studies would allow more robust conclusions. Additionally, there is a lack of data on male sling and AUS continence success rates in patients who have undergone radiation or brachytherapy. There have been reports of improved continence with the male sling after prostate radiation, but these were not compared with AUS [25]. Additionally, responses to some of these reports suggest that the AUS is a more appropriate intervention given that slings rely on redundant spongiosum for support, which may be compromised during radiation treatment [26]. In our practice, we have seen lower success rates with slings and worry about increased reoperation rates or conversion to AUS due to known fibrotic changes associated with radiation therapy. This is largely based on the diminished ability to provide urethral support, which the sling relies on to reestablish continence. While the various methods of prostate cancer ablation/treatment continue to evolve, the effects of these two specific modalities certainly impact the rates of incontinence. The studies included in our review excluded patients with moderate PPI who had undergone radiation therapy, but it would be reasonable to reassess these exclusions with the newer, more localized prostate cancer treatments now available. Finally, adjustable male slings were not included in this review as they are currently not available in the United States. While data for adjustable male slings appear to be non-inferior to fixed slings, the review articles summarized in Table 1 included fixed slings and therefore, studies conducted on adjustable slings were not included. As such, these limitations should be acknowledged when considering the overall argument of our presentation.

## 9. Future Directions

The male sling was the last advancement over 15 years ago, but there are two new treatments for PPI on the horizon, the Adjustable Continence Therapy (ProACT) and electronic AUS (eAUS). ProACT has been used in Europe since the early 2000s [27,28,29]. It was approved by the FDA in 2015 and has gained traction in some practices across the country [30]. The eAUS is still in the developmental phase, with an expected release in the next 5–10 years. The first reported eAUS was in November 2022 by the French Company UroMEMs [31,32]. It is not unreasonable to expect other companies to pursue the production of similar devices as this technology could be revolutionary for the field.

As discussed in the limitations section, our argument could be further strengthened with prospective RCTs that compare the continence rates of male slings versus AUS. These future studies would also benefit from longer follow-up periods and more objective outcome measures, as discussed in the “assessing incontinence” section. Additional research could be conducted to investigate the impact of radiation and brachytherapy on patients who are being considered for male slings or AUS. While these interventions tend to exclude patients from consideration, this criterion should be reevaluated to incorporate newer methods of prostate cancer treatment. The studies included in this article only utilized fixed slings, but the field would benefit from a review of adjustable slings versus fixed slings versus AUS.

## 10. ProACT

The adjustable continence therapy (ProACT) system consists of two silicon balloons that are inserted trans-perineally into the space where the prostate previously resided. The balloons are filled with an isotonic fluid at the discretion of the urologist and can be adjusted to apply pressure to the bladder neck. It holds the most promise for patients who present with PPI and are hesitant to undergo a more invasive and patient-controlled procedure such as an AUS. The mechanism of ProACT is similar to the sling whereby additional urethral support is achieved. In 2019, Larson conducted the first systematic review and meta-analysis for ProACT. Their review included 1264 patients in 19 studies and found that 60.2% of all patients with ProACT were able to achieve a “dry” status and 81.9% either achieved “dry” status or had a 50% improvement in their symptoms. Patients were using an average of 4.0 PPD at baseline, which was reduced to 1.1 PPD after ProACT implantation (CI 0.5–1.7). Bladder or urethral perforation was the most common complication, found in 5.3% of all cases, which occurred during the implantation of ProACT [33]. The authors believe that these complications will decrease over time as surgeons gain more experience. At this time, there are no long-term data assessing the use of ProACT in PPI patients, which may have resulted in short-term overestimates of continence and underestimates of the complication rates. While the rates of revision for any reason were found to be 22%, most of these procedures can be performed in the office setting with topical anesthesia. Overall, the authors felt ProACT should be considered as a first-line treatment for non-irradiated patients with PPI who are not appropriate candidates for AUS at the time of presentation, which is consistent with the exclusion criteria for slings. It should be noted that the data analysis for this study was supplied by UroMedica, Inc. (Plymouth, MN, USA), the developers of ProACT.

## 11. Electronic AUS

As discussed before, eAUS is still in the developmental phase but could be a monumental technological advancement for the field, as it eliminates many of the issues that the manual AUS presents (i.e., cognitive deficits and dexterity concerns). With the elimination of the scrotal pump, eAUS may require less patient interaction and possibly reduce the risk of device malfunction and erosion. Finally, pressures could be quantified with a target range, allowing the user to have greater control over their pump compared to the manual AUS. While we are still several years from public release, the eAUS may be the most revolutionary step in male urinary incontinence since the artificial sphincter itself.

## 12. Patient Hesitancy and Values

When the AdVance sling was introduced by Boston Scientific in 2007, a new discussion between patients and physicians surfaced regarding post-prostatectomy incontinence: which treatment is superior, the sling or AUS [34]? These conversations continue today and can leave patients hesitant to undergo a more invasive approach like an AUS when a seemingly simpler procedure such as the sling is available. In these situations, the conversation between patients and physicians is instrumental in determining which intervention is best based on patient goals and preferences. As demonstrated in studies since 2007, patients typically proceed with whichever intervention is specifically recommended by their physician. When given an opportunity to select either the sling or AUS, 92% of patients choose the sling, likely due to avoidance of an implantable mechanical device. When the physician recommends AUS, 75% of patients proceed with that recommendation [21]. These data suggest that physician perspective is one of the most important elements in treatment selection.

While functional continence or “dryness” has been the focus of this review, it is important to note that additional factors may sway patient decisions, like the physician’s role in patient-centered decision making. In recent studies, Hampson and Shaw both demonstrated that patients may value the simplicity of a proposed treatment, the possible need for future interventions, and prior patient experiences as much as continence itself [35,36]. While these values may be difficult to explicitly ascertain, it is important that physicians gain insight into patients’ complete perspective. A more invasive procedure, like an AUS, may initially seem daunting to a patient, but we have seen these reservations give way to acceptance once patients have a full understanding of what each treatment entails and their associated outcomes and complications. Patient fear of surgery should always be taken seriously, but suitable candidates should be informed that up to 35% of patients report regret after pursuing conservative therapy instead of a sling or AUS [37]. In summary, it is the physician’s role to explain and recommend treatments based not solely on the clinical outcome of continence, but also on additional factors that commonly concern all patients.

## 13. Conclusions

Regardless of the methodology used, the existing body of literature demonstrates AUS to be a more effective treatment than the male sling for men with moderate post-prostatectomy incontinence (>2 PPD or ≥Grade 2 MSIGS). While many patients are hesitant to undergo a more invasive procedure such as an AUS, it behooves the surgeon to discuss the current research and convey the reality that AUS provides better overall management of PPI and greater day-to-day satisfaction. Fears over the need for reintervention with an AUS are understandable considering the risk of device malfunction, infection, and erosion. The existing body of literature, however, has proven that patients who undergo a sling procedure can expect a second surgery at a similar or higher rate than patients with an AUS due largely to the high likelihood of persistent incontinence in the moderate PPI cohort. This is critical when assessing the risks and benefits of both interventions in patients with moderate PPI. While neither AUS nor the male sling represents a “cure” for incontinence, the AUS has proven to be the most effective procedure for reducing moderate post-prostatectomy incontinence and allows the patient to achieve social continence (0–1 PPD). Physicians should discuss the surgical options with the patient in the early post-operative period and strongly recommend early pelvic floor physical therapy (PFPT). PPI is a complex disease that requires a multimodal approach. We recommend that patients be referred to providers who are able to guide the patient through the entire PPI treatment pathway from PFPT to either a sling or AUS early in their disease course. We believe this will facilitate shared decision making with the patient when surgery is eventually discussed, rather than waiting 6–12 months after their prostatectomy. Furthermore, we have proposed an algorithm to help simplify the management of PPI that guides patients and physicians from pre-prostatectomy through the treatment of PPI (See Figure 1).

In our practice, we almost exclusively place an AUS for men with moderate PPI. Patients who present with PPI are always advised to initiate pelvic floor physical therapy if they have not already done so. While several benefit greatly from this approach alone, a subset fail conservative therapy and will require surgical intervention at 1 year. Despite this, we are diligent about maintaining patients on physical therapy even after an AUS is placed and have anecdotally found it to improve post-operative SUI, particularly in men who had excellent urinary control (0 PPD) in the immediate postop device-activation period. It would be interesting to compare patients with PPI who undergo AUS placement while continuing physical therapy with those who do not. The authors theorize that while the AUS provides compression, perhaps PFPT can increase the outlet resistance and thus improve upon the continence provided by the AUS. This may further improve the continence rates after AUS surgery and strengthen the argument for its use in moderate PPI by minimizing the risk of reintervention.

## Figures and Tables

**Figure 1 cancers-15-04424-f001:**
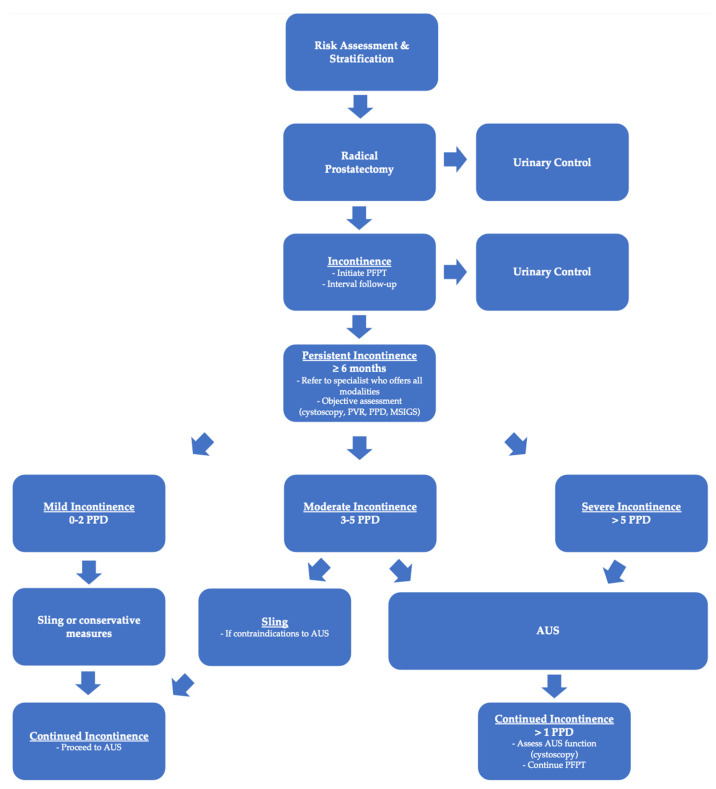
Proposed algorithm for assessment and management of patients with post-prostatectomy incontinence.

**Table 1 cancers-15-04424-t001:** Overview of selected studies.

Study	Incontinence Severity	Population	Outcome	Success Rates	Complications
Kourhi (2020) [8] *Retrospective*	MSIGS: Grade 2–3	179 total patients: 114 sling 65 AUS	0–1 PPD	AUS: 80% Sling: 63% *p* < 0.01	No difference
Sacco (2021) [9] *Retrospective*	Moderate: 3–5 PPD	70 total patients: 35 sling 35 AUS	“≥much improved” on survey AUS: −4 PPD Sling: −3 PPD	AUS: 94.5% Sling: 68.6%	Sling: 37% AUS: 20%
Abrams (2021) [10] "MASTER Trial" *Noninferiority RCT*	Moderate: 3 PPD 24 h pad weight: 256–267 g	380 total patients: 190 sling 190 AUS	Primary: No difference Secondary: survey results (see success)	Sling: more everyday life interference AUS: higher satisfaction with procedure	SAE’s: AUS (11) vs. sling (6) Future surgery: Sling (7.2%) vs. AUS (1.8%)
Lin (2022) [11] *Meta-analysis*	≤5 PPD	5 studies 509 total patients: 295 sling, 214 AUS	≤1 PPD	AUS: OR = 0.57 (CI 0.30–0.90)	No difference

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
