# Peer review of "Artificial Urinary Sphincters for Moderate Post-Prostatectomy Incontinence: Current Research and Proposed Approach"

_cancers, 2023, doi:10.3390/cancers15184424_

Round 1

Reviewer 1 Report

Dear Authors,

I read with interest your manuscript entitled: “Artificial Urinary Sphincters for Moderate Post-Prostatectomy Incontinence: Current Research and Proposed Approach”. 

The study is proposed as a non-systemic review of the literature on a topic that has gripped urologists for quite some time and on which there are few randomized controlled trials: stress urinary incontinence after radical prostatectomy. Not only that, but the authors also sought to evaluate the role of surgical treatment in moderate urinary incontinence after surgery. 

Although the review is ambitious there are some important limitations. 

First, the authors do not describe the type of review performed and how to search for papers regarding the topic. This could result in numerous errors since only the papers of interest may have been extrapolated to the exclusion of the counterpart. Therefore, I recommend adding a paragraph specifying and describing how this search was done and which papers were selected, perhaps listing it in a table.

The main missing element within this work is undoubtedly the specification on the type of sling and the type of artificial sphincter. Adjustable sling (e.g., ATOMS, VIRTUE) have been on the market for several years and have an indication precisely in that category of incontinence defined as moderate. Did the authors under the term SLING mean both adjustable and fixed sling?

In this paper, for example, which I recommend reading and adding to your review, the authors state that they treated mostly patients with moderate SUI, achieving a 79 percent social continence rate: 

·      Giammò A, Ammirati E, Tullio A, Morgia G, Sandri S, Introini C, Canepa G, Timossi L, Rossi C, Mozzi C, Carone R. Implantation of ATOMS® system for the treatment of postoperative male stress urinary incontinence: an Italian multicenter study. Minerva Urol Nephrol. 2020 Dec;72(6):770-777. doi: 10.23736/S0393-2249.19.03457-X. Epub 2019 Nov 4. PMID: 31692302.

Also in this work a high success rate at 36 months for the VIRTUE sling is made clear:

·      Roumeguère T, Elzevier H, Wagner L, Yiou R, Madurga-Patuel B, Everaert K, Chartier-Kastler E, Hegarty PK. The Virtue quadratic male sling for postradical prostatectomy urinary incontinence: 3-Year outcome measurements and a predictive model of surgical outcome from a European prospective observational study. Neurourol Urodyn. 2022 Jan;41(1):456-467. doi: 10.1002/nau.24851. Epub 2021 Dec 9. PMID: 34888939.

In this work, the modality of evaluation of urinary incontinence with subjective or objective parameters is also discussed. However, the number of diapers used and the MSIGS which can also be affected by the different abdominal pressure (just think of the bias of the BMI) do not represent safe and efficient methods.

The evaluation of urinary incontinence in patients post radical prostatectomy, should always combine objective quantitative and individual subjective parameters. Therefore, the role of the pad test in 24 hours, repeated for 3 days is certainly the safest objectifiable element, associated with the questionnaires evaluated. Therefore, in this regard, I recommend adding and citing this work to the review:

·      Mariotti G, Sciarra A, Salciccia S, Cattarino S, Fiori C, Gentilucci A. Quantitative analysis of urinary incontinence after prostatectomy: lack of standardization in trials. Minerva Urol Nephrol. 2022 Jun;74(3):249-251. doi: 10.23736/S2724-6051.22.04958-8. PMID: 35607781.

I advise the authors to make the requested changes and to develop a fairer discussion, to be able to give a more accurate indication to the readers.

Reviewer 2 Report

I commend the authors. A well thought out presentation. I have some thoughts/additions the authors at their discretion may consider.

A. A comment on avoiding PPI in the first place in addition to what has been included regarding MRI assessed MUL might be considered including adding issues such as surgeon skill at maximizing membranous urethral length as demonstrated via transparent PPI outcomes. 

B. Following Radical prostatectomy there is clear evidence that PPI outcomes are linked to both to surgeon experience and most importantly individual surgeon skill. Regardless, PPI continues especially in men > 75 years of age and in men with problematic "Overactive bladder".

Reviewer 3 Report

Dear authors,

I want to express my sincere gratitude for your hard work in putting together this much-needed review paper on the topic of AUS and male slings for the treatment of moderate post-prostatectomy incontinence. Your comprehensive analysis and synthesis of the current literature provide valuable insights into an important topic. As I move on to the specific aspects of the review, I would like to offer some constructive comments and suggestions that, I believe, will further enhance the quality and impact of your work.

Introduction: The introduction should provide more context and background information on the prevalence of post-prostatectomy incontinence, its impact on patients' quality of life, and the need for effective treatment options. Additionally, including a brief overview of the male sling and AUS procedures at the start of the introduction can help readers understand the relevance of the subsequent sections.

Methods and Inclusion Criteria: The article lacks information on the specific search strategy used to gather the studies included in the review. It is essential to outline the databases, keywords, and criteria used for study selection to ensure transparency and replicability.

Limitations: While the authors acknowledge some limitations, they should further emphasize the need for more high-quality, prospective, randomized controlled trials comparing AUS and slings for moderate PPI. The lack of such studies weakens the strength of the conclusions, and readers should be made aware of this limitation. Additionally, it would be beneficial to include the impact of radiation therapy on incontinence outcomes and the lack of data on using AUS or slings in patients who under goes radiation therapy as potential limitations.

Future Directions: The section on future directions is informative, but it should also include potential areas of research that could address the limitations of current studies, such as larger randomized controlled trials with longer follow-up periods and more objective outcome measures. Specifically, future studies should aim to investigate the impact of radiation therapy on the efficacy of AUS and slings in post-prostatectomy patients.

Conclusion: In the conclusion, summarize the main findings concisely and reiterate the recommendation for AUS as the preferred treatment for moderate PPI. Additionally, it would be essential to include a section about the lack of data on the impact of radiation therapy following AUS or male sling, and provide expert opinion on for managing incontinence in patients who would undergo radiation therapy. Also, emphasize the importance of shared decision-making and patient counseling in treatment selection, taking into account individual patient characteristics, including any history or potential need for radiation therapy in the future.

Once again, thank you for your valuable contribution to the field.

Round 2

Reviewer 1 Report

Authors correctly answered to reviewers comments